# Whole Exome Sequencing Is the Minimal Technological Approach in Probands Born to Consanguineous Couples

**DOI:** 10.3390/genes12070962

**Published:** 2021-06-24

**Authors:** Francesca Peluso, Stefano Giuseppe Caraffi, Roberta Zuntini, Gabriele Trimarchi, Ivan Ivanovski, Lara Valeri, Veronica Barbieri, Maria Marinelli, Alessia Pancaldi, Nives Melli, Claudia Cesario, Emanuele Agolini, Elena Cellini, Francesca Clementina Radio, Antonella Crisafi, Manuela Napoli, Renzo Guerrini, Marco Tartaglia, Antonio Novelli, Giancarlo Gargano, Orsetta Zuffardi, Livia Garavelli

**Affiliations:** 1Medical Genetics Unit, Azienda USL-IRCCS di Reggio Emilia, 42123 Reggio Emilia, Italy; francesca.peluso@ausl.re.it (F.P.); stefanogiuseppe.caraffi@ausl.re.it (S.G.C.); roberta.zuntini@ausl.re.it (R.Z.); gabriele.trimarchi@ausl.re.it (G.T.); ivanovski@medgen.uzh.ch (I.I.); lara.valeri@ausl.re.it (L.V.); Veronica.Barbieri2@ausl.re.it (V.B.); Maria.Marinelli@ausl.re.it (M.M.); 2Institut für Medizinische Genetik, Universität Zürich, 8952 Zürich, Switzerland; 3Post Graduate School of Paediatrics, University of Modena and Reggio Emilia, 41124 Modena, Italy; alessiapancaldi@gmail.com; 4Neonatal Intensive Care Unit, Azienda USL-IRCCS di Reggio Emilia, 42123 Reggio Emilia, Italy; Nives.Melli@ausl.re.it (N.M.); giancarlo.gargano@ausl.re.it (G.G.); 5Translational Cytogenomics Research Unit, Bambino Gesù Children’s Hospital, IRCCS, 00165 Rome, Italy; claudia.cesario@opbg.net (C.C.); emanuele.agolini@opbg.net (E.A.); antonio.novelli@opbg.net (A.N.); 6Pediatric Neurology, Neurogenetics and Neurobiology Unit and Laboratories, Meyer Children’s Hospital, University of Florence, 50139 Florence, Italy; elena.cellini@meyer.it (E.C.); renzo.guerrini@meyer.it (R.G.); 7Genetics and Rare Diseases Research Division, Ospedale Pediatrico Bambino Gesù, IRCCS, 00165 Rome, Italy; fclementina.radio@opbg.net (F.C.R.); marco.tartaglia@opbg.net (M.T.); 8Pediatric Unit, Azienda USL-IRCCS di Reggio Emilia, 42123 Reggio Emilia, Italy; Antonella.Crisafi@ausl.re.it; 9Neuroradiology Unit, Azienda USL-IRCCS di Reggio Emilia, 42123 Reggio Emilia, Italy; Manuela.Napoli@ausl.re.it; 10Unit of Medical Genetics, Department of Molecular Medicine, University of Pavia, 27100 Pavia, Italy; orsetta.zuffardi@unipv.it

**Keywords:** *KATNB1*, lissencephaly 6, microcephaly, *FAT1*, microphthalmia, preaxial polydactyly, split foot, next generation sequencing, consanguinity

## Abstract

We report on two siblings suffering from different pathogenic conditions, born to consanguineous parents. A multigene panel for brain malformations and microcephaly identified the homozygous splicing variant NM_005886.3:c.1416+1del in the *KATNB1* gene in the older sister. On the other hand, exome sequencing revealed the homozygous frameshift variant NM_005245.4:c.9729del in the *FAT1* gene in the younger sister, who had a more complex phenotype: in addition to bilateral anophthalmia and heart defects, she showed a right split foot with 4 toes, 5 metacarpals, second toe duplication and preaxial polydactyly on the right hand. These features have been never reported before in patients with pathogenic *FAT1* variants and support the role of this gene in the development of limb buds. Notably, each parent was heterozygous for both of these variants, which were ultra-rare and rare, respectively. This study raises awareness about the value of using whole exome/genome sequencing rather than targeted gene panels when testing affected offspring born to consanguineous couples. In this way, exomic data from the parents are also made available for carrier screening, to identify heterozygous pathogenetic and likely pathogenetic variants in genes responsible for other recessive conditions, which may pose a risk for subsequent pregnancies.

## 1. Introduction

Consanguineous marriages reach peaks of 20–50% in various regions of the Middle East and the Mediterranean basin [1,2], while in other parts of the world, inbreeding is essentially the remain of the *latifundia* economy, and current consanguineous couples are often unaware that they have common ancestors [3].

The risk of genetic diseases in the offspring of consanguineous couples is significantly higher (6–10%) than in unrelated unions, even when family history is negative [4,5,6]. In a union between first cousins, if one parent has a very rare recessive variant, the possibility that the other parent is also a carrier shifts from the Hardy–Weinberg equilibrium, changing the probability from negligible to very high. In Saudi Arabia, a premarital screening program of 503 couples estimated that >12% was at reproductive risk due to consanguinity, even counting only the set of pathogenic variants previously detected in that population [7]. Compared to the risk of 1.2% for couples in Western countries, the difference is dramatic, and Europe is therefore called up on to provide dedicated health programs for the many migrant couples from countries where intermarriages are a common practice. 

Considering that the extension of homozygous regions (ROH) increases with the degree of parental consanguinity, the risk of offspring with diverse recessive diseases due to variants in different genes must be taken into account, especially in the children of first-cousin couples.

In this study, we present the case of a first-cousin couple who had two children with severe malformation syndromes caused by homozygous variants in two different genes: *KATNB1* and *FAT1* We provide a comprehensive clinical case report of the two affected sibs, and provide new data expanding the clinical features associated with pathogenic variants in *FAT1*, a gene better known for the role of its somatic mutations in cancer than its germinal variants in Mendelian disorders.

## 2. Materials and Methods

The probands and their parents have provided written informed consent for molecular analyzes. 

Genomic DNA was extracted from peripheral blood leucocytes with QIAgen columns (QIAsymphony DNA minikit, Qiagen, Hilden, Germany) according to the manufaturer’s instructions. Concentration and purity of DNA samples were quantified by ND-1000 spectrophotometer (NanoDrop; Thermo Scientific, Waltham, MA, USA) and by FLx800 Fluorescence Reader (BioTek, Winooski, VT, USA). 

Targeted resequencing of a panel including 83 genes associated with cerebral malformation was performed in the patient. Target enrichment and library preparation were performed using a custom-designed Nextera Rapid Capture assay (Illumina, San Diego, CA, USA), and sequencing was performed on an Illumina MiSeq (Illumina, San Diego, CA, USA) with a 2 × 150-bp paired-end protocol as previously described [8]. Variants were annotated and filtered using the ANNOVAR tool [9]. Variants localized in intronic regions outside the 10-bp exon-flanking boundaries and in the 5′- and 3′-UTR regions were excluded. Variants reported in the Exome Aggregation Consortium (ExAC) database (exac.broadinstitute.org/, accessed on 19 March 2021) and/or in the 1000 Genomes Project (http://www.international genome.org/, accessed on 14 September 2016), in the National Heart, Lung, and Blood Institute Exome Sequencing Project (ESP, ESP6500 database, EVS: evs.gs.washington.edu/EVS, accessed on 14 September 2016), in the Genome Aggregation Database (gnomAD, gnomad.broadinstitute.org/, accessed on 14 September 2016), with a minor allele frequency >0.01 (1%) were dropped out. In silico prediction of mutations’ pathogenicity was obtained using the dbNSFP database (v3.3), which provides functional prediction scores on more than 20 different algorithms (sites.google.com/site/jpopgen/dbNSFP, accessed on 14 September 2016). Putative causative variants were analyzed by Sanger sequencing to confirm the next-generation sequencing (NGS) results in probands, and were investigated in the parents to check the inheritance status. We classified variants according to the international guidelines of the American College of Medical Genetics and Genomics (ACMG) Laboratory Practice Committee Working Group [10]

Trio-based whole-exome sequencing (WES) was performed on genomic DNA by using the Twist Human Core Exome Kit (Twist Bioscience, South San Francisco, CA, USA) according to the manufacture’s protocol on a NovaSeq6000 platform (Illumina, San Diego, CA, USA). The reads were aligned to human genome build GRCh37/UCSC hg19. The Dragen Germline Enrichment application of BaseSpace (Illumina, San Diego, CA, USA) and the Geneyx Analysis (knowledge-driven NGS Analysis tool powered by the GeneCards Suite) were used for the variant calling and annotating variants, respectively. Sequence data were carefully analyzed and the presence of all suspected variants was checked in the public databases (dbSNP (http://www.ncbi.nlm.nih.gov/projects/SNP, accessed on 25 September 2020), 1000 Genomes Project, EVS, ExAC, gnomAD). Exome sequencing data filtering was performed to identify protein-altering, putative rare recessive homozygous, compound heterozygous, and pathogenic or likely pathogenic heterozygous variants with an allele frequency <1%, according to ExAC’s overall frequency, which result in a change in the amino acid sequence (i.e., missense/nonsense), or that reside within a canonical splice site. The variants were evaluated by VarSome [11] and categorized in accordance with the ACMG recommendations [10]. Variants were examined for coverage and Qscore (minimum threshold of 30) and visualized by the Integrative Genome Viewer (IGV). Metrics of the WES data output are summarized in the Appendix A.

Further in silico prediction tools and databases used for variant analysis include the Human Gene Mutation Database (HGMD; https://portal.biobase-international.com/, accessed on 25 September 2020), MaxEntScan (http://hollywood.mit.edu/burgelab/maxent, accessed on 25 September 2020), Berkeley Drosophila Genome Project Splice Site Prediction by Neural Network (BDGP-SSP: https://www.fruitfly.org/seq_tools/splice.html, accessed on 25 September 2020), and NMD Escape Predictor (https://nmdprediction.shinyapps.io/nmdescpredictor/, accessed on 25 September 2020), SNAP2 (https://rostlab.org/services/snap2web/, accessed on 25 September 2020).

## 3. Patients and Results

### 3.1. Case Reports

The parents, who were apparently healthy and had not been exposed to teratogens, were first cousins from Punjab (India) (Appendix A). They had four daughters, the second and fourth born with malformations. At their births, the mother was 30 and 36 years old and the father 35 and 41, respectively. A stillbirth with unspecified brain malformations has also been reported. 

### 3.2. Case 1

The mother reported normal fetal movements. The patient was born at 37 weeks of gestation, with the following parameters: weight 2090 g (<3rd percentile, −2.06 SD), length 44 cm (<3rd percentile, −1.98 SD), head circumference 31 cm (<3rd percentile, −1.83 SD), Apgar score 1′: 9, 5′: 10. After birth, she was admitted to the hospital due to jaundice and tachypnea. Psychomotor development was delayed: sitting without support at 9 months, autonomous walking at 22 months, first words at 3 years and 10 months. She showed hyperactivity with episodes of self-aggression, and no sleep disturbances.

Somatometric data at birth and from subsequent clinical examinations, up to the age of 3 years and 10 months, are summarized in Figure 1M. 

She had primary microcephaly, slight posterior plagiocephaly, bilateral ptosis, nystagmus, anteverted nostrils and down-turned corners of the mouth (Figure 1A,B). 

Hands: single palmar crease on the left hand, slight bilateral clinodactyly of the 5th finger. Feet: partial two–three toe syndactyly on the right foot (Figure 1C–F).

Blood investigations: carnitine, acyl-carnitine, 7-dehydro-cholesterol, all normal. Heart ultrasound: subaortic ventricular septal defect (VSD) (diameter 3 mm) with left-right shunt and atrial septal defect (ASD). Ophthalmologic evaluation: bilateral ptosis, nystagmus, retinal pigmentation. Audiometric evaluation (ABR): slight hearing loss on the left side. Brain MRI (age 3 months): subarachnoid dilatation and abnormal gyration in the frontotemporal regions, slight hippocampal malrotation, subcortical heterotopia, slightly thick cerebral cortex (Figure 1G–L). 

An NGS panel including 83 genes associated with brain malformations and microcephaly showed an homozygous splice site variant in *KATNB1*: NM_005886.3:c.[1416 + 1del]; [1416 + 1del]. This variant, confirmed by Sanger sequencing, was not reported in gnomAD v2.1.1 or HGMD (accession on 7 April 2021) and was predicted to disrupt proper transcript processing (MaxEntScan, BDGP). The variant was present in both parents in the heterozygous state. No other variants of potential interest were found in the NGS panel. The novel variant in *KATNB1* has been submitted to the Leiden Open Variation Database (ID 0000789679).

### 3.3. Case 2

Pregnancy was complicated by gestational diabetes. Prenatal ultrasounds were normal and fetal movements were present. She was born at 39 weeks of gestation, with the following parameters: weight 2780 g (7th percentile, −1.36 SD), length 50 cm (50th percentile, 0.12 SD), head circumference 33. 5 cm (25th percentile, −0.57 SD), Apgar score: 1′: 9, 5′: 10. At birth she was admitted to the Neonatal Department because of severe bilateral microphthalmia (Figure 2A–E), congenital heart defect (subaortic VSD, ASD *ostium secundum*, bicuspid aortic valve), right split foot with four toes and three–four syndactyly, left foot with polydactyly affecting the second toe and nail (Figure 2I–K).

Psychomotor development was characterized by sitting without support at 10 months, while language development (babbling) was normal. Clinical examination at the age of 10 months: length 67 cm (4rd percentile, −1.78 SD), weight 6900 g (3rd percentile, −2.47 SD), head circumference 42.3 cm (3rd percentile, −1.87 SD) (Figure 2U). Due to a nephrotic syndrome with high levels of proteinuria (190 mg/dl, normal range: 0–25) and hypoproteinemia (4 g/dl, normal range: 5.7–8.2), she started treatment with albumin infusions. Blood and urine tests including complete blood count and gas, blood urea nitrogen, creatinine, transaminase, γ GT, serum electrolytes, albumin, urinary creatinine were all normal. Serological testing detected low levels of IgG (131 mg/dl, normal range for age 279–1533), but normal values of IgM (75 mg/dl, normal range for age 22–147) and IgA (20.3 mg/dl, normal range for age 16–98).

The following examinations were also performed: transfontanellar ultrasound: normal; echocardiography: subaortic VSD, ASD *ostium secundum*, bicuspid aortic valve; abdominal ultrasound: normal; audiometric evaluation: bilateral sensorineural hearing loss of medium gravity for frequencies 2–4 kHz; ophthalmological evaluation: severe bilateral microphthalmia with bilateral coloboma; right hand X-ray: preaxial polydactyly with one metacarpal and two proximal and distal phalanges; right foot X-ray: split foot with four toes and five metatarsals: curved second metatarsal; left foot X-ray: second toe polydactyly; brain MRI: no abnormal gyration, no thickened cerebral cortex, no hippocampal malrotation. Normal signal and morphology of the brain nervous tissue. The ventricular system appears in place, with regular morphology and size. Bilateral enophthalmos with markedly dysmorphic appearance of the eyeballs, with profile irregularities, in particular on the posterior part in correspondence with the head region of the optic nerve and bilateral coloboma. On the left, dysmorphic appearance and posterior displacement of the lens, anchored to the ciliary body/suspensory ligaments that appear stretched with secondary enlargement of the anterior chamber; on the posterior side the lens appears deformed and attracted posteriorly in relation to the persistence of the hyaloid canal (persistent hyperplastic primary vitreous body, PHPV). A more nuanced and slender PHPV finding is also evident in the right eyeball where the posterior surface of the lens appears slightly dysmorphic. The optic nerves had a slightly tortuous course in the retrobulbar tract while the chiasm and optic tracts were normal. 

Microarray revealed a partial duplication of the long arm of a chromosome 15 (15q26.3) which extends for about 350 Kb (GRCh37, chr15: 100, 755, 18001, 105, 284 dup) and includes the OMIM morbid gene *CERS3* and part of the OMIM morbid gene *ADAMTS17*. The duplication was inherited from the healthy mother and it had no consistent relation with the proband’s phenotype.

Based on the complex clinical phenotype, the patient was enrolled in the Italian network for undiagnosed patients. Trio-based whole exome sequencing (WES) revealed a homozygous variant in the *FAT1* gene: NM_005245.4: c.[9729del]; [9729del], NP_005236.2:p.[(p.Val3245Leufs*25)]; [(p.Val3245Leufs*25)]. It determines a frameshift and the introduction of a premature stop codon, which is likely to lead to nonsense-mediated mRNA decay (NMD Escape Predictor). Even if translated, it would generate a truncated protein lacking about 1/4th of its sequence, including the transmembrane, the laminin-G like and all the EGF-like domains. This variant, confirmed via IGV, was segregated from both heterozygous parents and has been reported in public databases with a frequency of 1 out of 249194 alleles (gnomAD v2.1.1, accession on 7 April 2021 dbSNP: rs1445858471) and submitted in HGMD with accession ID CD193777 as a homozygous mutation in two brothers with intellectual disability and bilateral syndactyly of the feet, from a consanguineous Pakistani family. Search for this variant was not performed in the healthy sisters. None of the *de novo* or biallelic VUS detected in WES were involved in biological pathways capable of explaining the patient’s features (Appendix A).

## 4. Discussion

We describe two sisters (cases 1 and 2) born to consanguineous parents, the first presenting essentially with microcephaly, brain malformations and severe psychomotor retardation, the second with microphthalmia, coloboma, cardiac anomalies, renal dysfunction and limb malformations (Table 1). 

The homozygous variants, c.1416 + 1del in the *KATNB1* gene (16q21) of case 1 and c.9729del in the *FAT1* gene (4q35.2) of case 2, are both predicted to result in a loss-of-function, conform to the malformation characteristics of the respective patients and can be classified as “likely pathogenic” according to the ACMG/AMP criteria [10]. 

*KATNB1* gene encodes one of the two subunits of the microtubule-severing protein complex katanin. As expected for variants in genes involved in microtubules and microtubule-associated proteins, its variants are enriched in syndromes characterized by microcephaly and abnormal cortical growth and gyrification [12]. Indeed, interactions between the *KATNB1* mutant with *KATNA1*, the Katanin catalytic subunit, and other microtubule-associated proteins result in defective mitotic spindle in patient-derived fibroblasts [13]. Furthermore, *katnb1* null mutant mouse embryos revealed the role of this gene in regulating cilia number and function within the hedgehog signaling pathway, [12,14].

So far, homozygous and compound heterozygous mutations in *KATNB1* have been reported in 13 subjects, aged 11 months to 12 years, from nine families [12,13,15] (Appendix A). 

Microcephaly is a characteristic feature of *KATNB1*-related syndrome, with head circumference between −1.63 and −5.9 SD at birth, while weight and length at birth are normal or just below the normal limits. Subsequently a further reduction of the head circumference has been observed, in eight cases out of 14 down to less than −6 SD and in one case less than −11 SD by the age of 12 years. In the six patients with unavailable auxological data, growth retardation was noted.

Psychomotor development was regular in two patients, with a large degree of variability in the others, with five patients never achieving independent walking and five patients with absent speech. Common phenotypic features are synophrys, hypertelorism, anteverted nares, prominent nose and large ears. The main neuroimaging features are pachygyria, polymicrogyria, simplified gyral pattern, periventricular heterotopias, and bilateral nodular heterotopia of gray matter in the irradiated corona. Abnormalities of the corpus callosum are common, ranging from agenesis, either partial or complete, to shortening. Posteriorly enlarged ventricles, enlarged cisterna magna, Dandy–Walker variant, and cystic enlargement of the fourth ventricle was reported in a few cases. Neurological findings showed hypertonia mainly in the lower limbs in eight patients out of 14. 

Eye anomalies were also reported and included nystagmus, strabismus, pigmented retina and optic nerve atrophy. Case 1 has bilateral eyelid ptosis, a previously unreported sign. Anomalies affecting the extremities were broad thumbs, clinodactyly, tapering fingers, preaxial polysyndactyly, pes planus-valgus and two–three toe syndactyly, and the latter was also present in case 1.

Congenital heart defects are rare: case 1 has both a subaortic VSD and an atrial septal defect (ASD), the latter is also present in another patient. Uncommon features include extrahepatic biliary atresia, hypoplastic sternum, sacral dimple, and omphalocele.

The second sister, case 2, presents a malformative picture that is largely similar to the one reported for homozygous frameshift mutations in the *FAT1* gene [16], including colobomatous microphthalmia, ptosis, nephropathy and toe syndactyly, but also polydactyly of hand and foot and split foot.

*FAT1* is a member of the FAT cadherins, one of the gene families involved in basic developmental processes such as cell-to-cell adhesion. While *FAT1* somatic variants are very frequent in every type of human cancer [17], constitutive variants have only recently been associated with congenital conditions, with a recessive mode of inheritance. So far, biallelic variants in *FAT1* have been reported in 17 affected subjects, aged 9 months to 39 years, from 11 families [16,18,19,20,21] (Appendix A). 

Gene constraints in gnomAD v2.1.1 suggest that *FAT1* has a reduced tolerance to loss-of-function mutations. Although nonsense, frameshift and splicing variants have been identified across the entire gene in reference populations, they never occur in homozygosity. In fact, rare biallelic loss-of-function variants have been associated with a multisystem malformative syndrome, with coloboma being the most consistent developmental defect [16,18,20,21]. Asymmetric syndactyly of the toes is common, primarily affecting the third and fourth toes, and can be unilateral or bilateral, cutaneous or osseous. Hypotrophy of the phalanx of one foot has also been described. In case 2, malformation of the limbs was more complex, though still asymmetrical, and included not only syndactyly but also polydactyly in one hand and unilateral split foot. Altogether, these findings strongly indicate that this gene is a further interactor for the formation of limb buds, even if the frameshift variants are only partially penetrant, given the asymmetry of the malformations. Recent studies on squamous cell carcinoma of mice and humans have revealed that the loss of function of *FAT1* leads to an epithelial-mesenchymal hybrid transition phenotype, which, in turn, promotes tumor initiation, progression, invasiveness, stemness and metastasis [17]. Consequently, an abnormal reprogramming of gene expression during the localized epithelial-mesenchymal transition which is essential in the formation of limb progenitors [22] could explain the foot and hand abnormalities associated with *FAT1* biallelic frameshift variants. 

It is noteworthy that several DECIPHER patients with polydactyly and other hand, foot, ulna, radius, or metacarpal bone anomalies are associated with heterozygous deletions involving *FAT1*, some inherited from a healthy parent and others de novo (cases 263566, 263566, 249192, 276704, 273754, 285956, 300330, 384020, 392005, 392006, 392058, 392082, 392076, 392066, 393352, 392845, 394362, 394338, 394314, 394279, 395652, 401296, 398400, 398396). This observation suggests the presence in the non-deleted allele of medium frequency regulatory variants that lower protein production below the minimum level for normal limb embryogenesis, according to the TAR syndrome model [23].

In contrast, gene constraints in gnomAD v2.1.1 indicate tolerance for missense variants, as can be expected of a modular protein that consists of several repeats of a short immunoglobulin-like fold for most of its length. Accordingly, only a few biallelic *FAT1* missense variants in different cadherin ectodomains have been associated with a condition characterized by isolated glomerulotubular nephropathy [18,19]. Proteinuria, hematuria and/or glomerulotubular defects have also been identified in patients with the syndromic disease caused by frameshift variants, but only in half of all reported individuals (Appendix A). Taken together, these observations suggest that *FAT1* alterations may be associated with renal defects with reduced penetrance. In the case of missense variants, some are likely to be hypomorphic [18], reaching clinical significance only in rare individuals. The number of described patients with *FAT1*-related clinical manifestations is still too small, however, and several aspects of the genotype-phenotype correlations still need to be elucidated. We cannot exclude that some biallelic missense variants, leading to a severe functional reduction, may have variable penetrance for some of the dysmorphic features currently associated only with the frameshift defects. Furthermore, some missense variants have been demonstrated to cause splicing defects that mostly result in in-frame deletions within the cadherin ectodomains, and their heterozygous presence may predispose to a muscular dystrophy-like phenotype [24].

In this study, we describe an exceptional condition, namely two sisters suffering from two different rare diseases. Indeed, in consanguineous families, similar events might be expected to occur with a greater frequency than appears in the literature. Retrospective analysis of data from a series of consecutive unrelated patients sent to a clinical diagnostic laboratory for WES supports this eventuality [25]. A molecular diagnosis was made for 2076 out of 7374 patients; among these, 101 (4.9%) had diagnoses involving two or more disease loci. Two different autosomal genes with homozygous variants contributing to the phenotypic alterations of the proband were present in nine cases, indirectly suggesting that in consanguineous couples the probability of having offspring affected by two different recessive diseases, either present in the same patient or in two children, is at least 0.4%. 

From a clinical perspective, the diagnostic workflow may often be complicated for the offspring of consanguineous couples because some gene-disease associations are so rare that they occur only among consanguineous populations, remaining unknown in the rest of the world. 

In the family described here, the two sisters presented phenotypic pictures that were scarcely reminiscent of commonly known syndromes, even though, in retrospect, the identification of homozygous variants in *KATNB1* and *FAT1* well explained the respective malformations. In case 1, primary microcephaly and brain malformations, including in particular nodular heterotopia, severe short stature and syndactyly, are perfectly compatible with the homozygous *KATNB1* mutation we detected [15,26]. In case 2, colobomatous-microphthalmia, ptosis, nephropathy and limb malformation correspond dramatically to what is described in the few patients with *FAT1* homozygous frameshift variants reported so far [16], and strengthen the role of the novel variant we have identified. 

However, even before knowing the molecular data, the malformation features of the two sisters were so diverse that the hypothesis of a single pathogenic gene variant with variable expressivity was implausible. In this situation, the only solution is the sequencing of the “agnostic” genome of the whole family, which confirms the causality of the homozygous variant in the affected offspring through the absence of the variant or its heterozygous state in healthy individuals. On the other hand, when the proband shows well-defined characteristics such as microcephaly and brain malformations in case 1, the use of gene panels linked to the malformation and continuously updated is often preferred, especially in those centers where the WES approach is not routinely applied. 

In the present study, the two affected children were analyzed at different times and the family history could only be reconstructed retrospectively. Molecular analysis performed using a genetic panel in one case and WES in the other identified causal variants in two different genes and explained the phenotypic differences of the two girls, demonstrating the importance of addressing molecular diagnostics in consanguineous families by analyzing the whole set of genes in the entire family and the inappropriateness of the analysis through genetic panels designed for specific malformation syndromes. Furthermore, in healthy consanguineous parents, a whole exome/genome approach for carrier screening for identical heterozygous pathogenetic or likely pathogenetic variants (Class 4 and 5 according to ACMG/AMP classification) [10] related to autosomal recessive diseases is a powerful tool for defining the recurrence risk, facilitating informed reproductive decision-making and offering the possibility of prenatal and preimplantation genetic diagnosis in further pregnancies. 

In fact, considering those fetuses that do not have the disease-variant identified in the previous affected child as not at risk precludes the detection of homozygous variants in other disease genes. 

## 5. Conclusions

As for Europe, the dramatic conditions of poverty and war have created important migratory flows from the Near East. A situation of fragility such as that of migrant families cannot be further ignored and requires easier access to preconceptional carrier tests that examine the entire exome of the couple. It will be the task of the clinical geneticist to explain that knowing the autosomal recessive diseases for which their offspring is at risk does not exclude pathological conditions from *de novo* variants that occur, regardless of inbreeding, with a risk equal to that of the general population.

## Figures and Tables

**Figure 1 genes-12-00962-f001:**
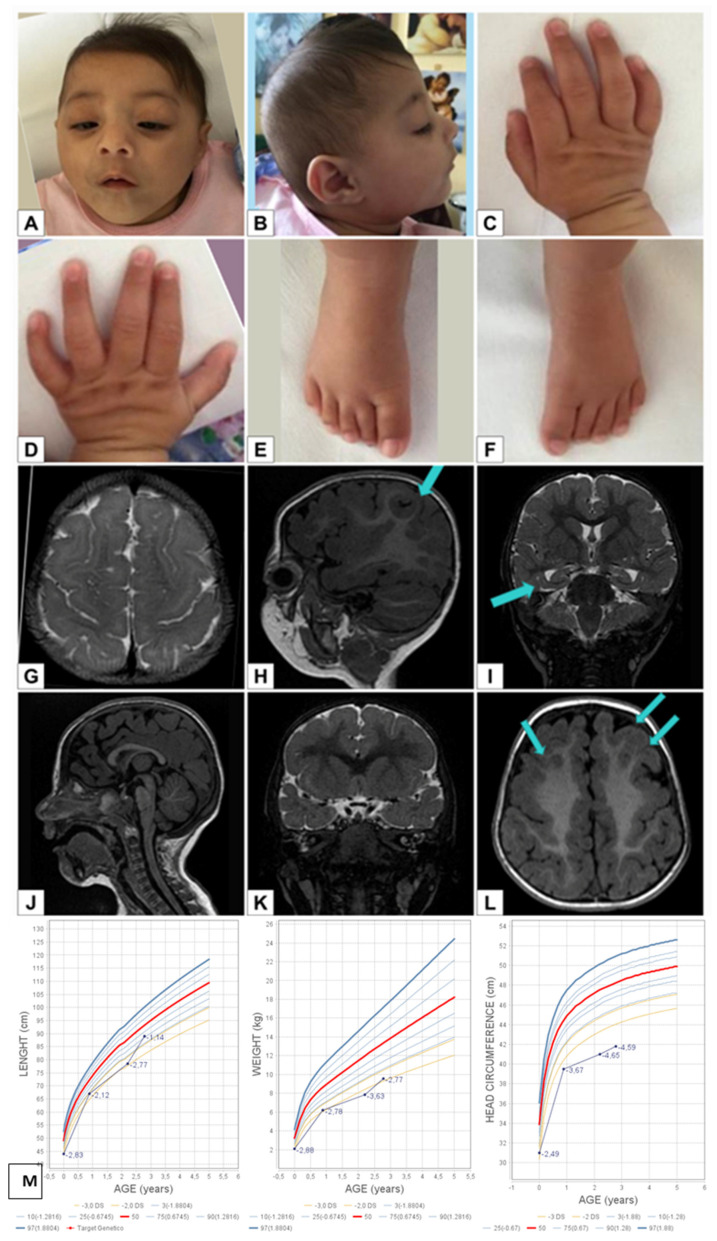
(**A**,**B**) Microcephaly, slight posterior plagiocephaly, bilateral ptosis, nystagmus, anteverted nostrils and down-turned corners of the mouth. (**C**–**F**) Hands: single palmar crease on the left hand, slight bilateral clinodactyly of the 5th finger. Feet: slight cutaneous syndactyly of second–third toes on the right foot. (**G**–**L**) Brain MRI (age 3 months): subarachnoid dilatation and abnormal gyration in the fronto-temporal regions, slight hippocampal malrotation, subcortical heterotopia, slightly thickened cerebral cortex. (**M**) Somatometric data at birth, 11 months, 2 years and 3 months, 3 years and 10 months of age.

**Figure 2 genes-12-00962-f002:**
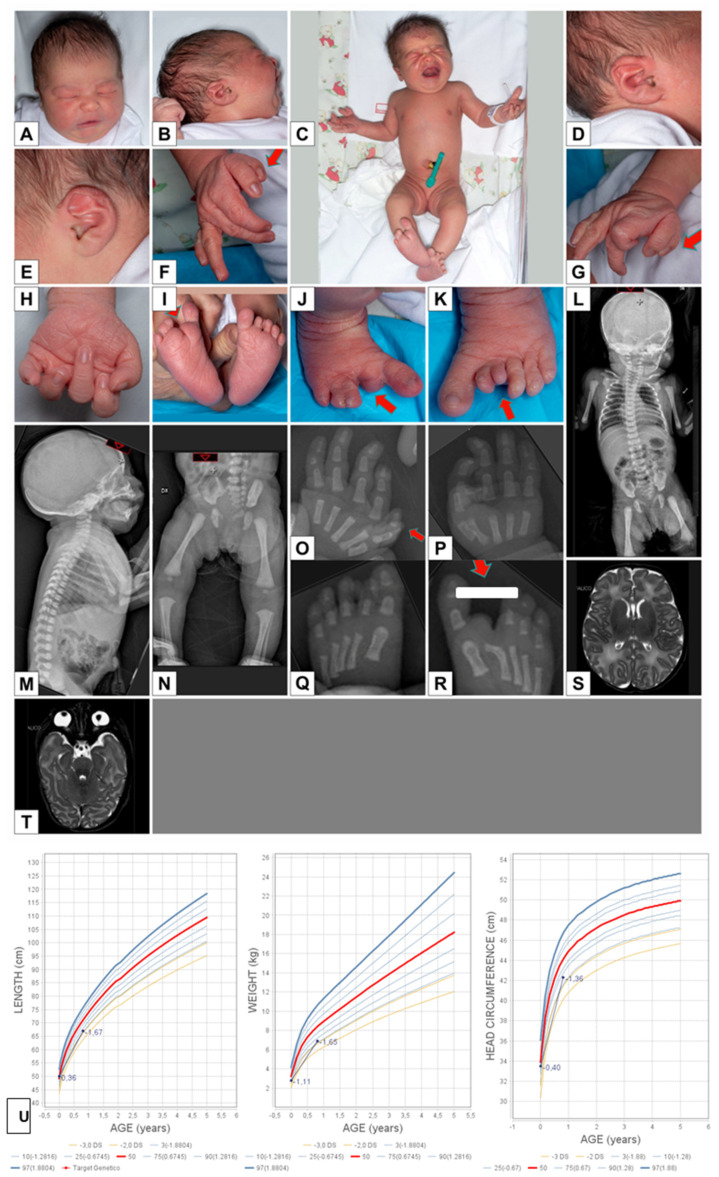
(**A**–**E**) Clinical examination at birth: normal head, severe bilateral microphthalmia. (**F**–**H**) Preaxial polydactyly on the right hand. (**I**–**K**) Right split foot with four toes and third–fourth syndactyly and on the left foot: second toe polydactyly with nail. (**L**–**N**) Skeletal X-ray: no vertebral anomalies, no long bone anomalies. (**O**–**R**) Right hand X-ray: preaxial polydactyly with one metacarpal and two proximal and distal phalanges; right foot X-ray: split foot with four toes and five metatarsals: curved second metatarsal; left foot X-ray: second toe polydactyly. (**S**,**T**) Brain MRI: Bilateral enophthalmos with markedly dysmorphic appearance of the eyeballs with profile irregularities in particular on the posterior part in correspondence with the head region of the optic nerve and bilateral coloboma. On the left, dysmorphic appearance and posterior displacement of the lens, anchored to the ciliary body/suspensory ligaments that appear stretched with secondary enlargement of the anterior chamber; on the posterior side the lens appears deformed and attracted posteriorly in relation to the persistence of the hyaloid canal (persistent hyperplastic primary vitreous body, PHPV). More nuanced and slender PHPV finding is also evident in the right eyeball where the posterior surface of the lens appears slightly dysmorphic. The optic nerves (with a slightly tortuous course in the retrobulbar tract) are appreciable bilaterally, the chiasm and the optic tracts are also regularly displayed. (**U**) Somatometric data at birth and at 10 months of life.

**Table 1 genes-12-00962-t001:** HPOs terms to resume phenotypes of case 1 and case 2.

	CASE 1—Homozygous c.1416 + 1del in *KATNB1*		CASE 2—Homozygous c.9729del in *FAT1*	
	HPO Number	Phenotype	HPO Number	Phenotype
**HEAD**	HP:0000252	Microcephaly		
	HP:0011327	Posterior plagiocephaly		
**MRI ANOMALIES**	HP:0012766	Widened cerebral subarachnoid space		
	HP:0002536	Abnormal cortical gyration		
	HP:0025100	Abnormal hippocampus morphology		
	HP:0032391	Subcortical heterotopia		
	HP:0006891	Thick cerebral cortex		
**EYES**	HP:0000666	Horizontal nystagmus		
	HP:0007703	Abnormality of retinal pigmentation		
	HP:0001488	Bilateral ptosis	HP:0001488	Bilateral ptosis
			HP:0007968	Remnants of the hyaloid vascular system
			HP:0007633	Bilateral microphthalmos
			HP:0000588	Optic nerve coloboma
**HEART**	HP:0011681	Subarterial ventricular septal defect	HP:0011681	Subarterial ventricular septal defect
	HP:0001684	Secundum atrial septal defect	HP:0001684	Secundum atrial septal defect
**SKELETON**			HP:0001177	Preaxial hand polydactyly
			HP:0001839	Split foot
			HP:0001829	Foot polydactyly
	HP:0004209	Clinodactyly of the 5th finger		
	HP:0005709	2–3 toe cutaneous syndactyly		
**DEVELOPMENT AND GROWTH**				
	HP:0001510	Growth delay		
	HP:0001263	Global developmental delay		

**KIDNEY**				
			HP:0000100	Nephrotic syndrome

**HEARING**			HP:0012716	Moderate conductive hearing impairment
	HP:0012712	Mild hearing impairment		

## Data Availability

The data that support the findings of this study are available from the corresponding author upon reasonable request.

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
