# Peer review of "Whole Exome Sequencing Is the Minimal Technological Approach in Probands Born to Consanguineous Couples"

_genes, 2021, doi:10.3390/genes12070962_

Round 1

Reviewer 1 Report

Whole exome sequencing is the minimal technological approach in probands born to consanguineous couples. Peluso et al. 2021

The authors reported here two genetic variants, KATNB1 and FAT1 gene, each from two siblings suffering from different pathogenic conditions. Two sisters born to consanguineous parents, however, two siblings showed different pathogenic phenotypes which is exceptional condition. The first old sister presents microcephaly, brain malformations and severe psychomotor retardation, the second with microphthalmia, coloboma, cardiac anomalies, renal dysfunction and limb malformations.

            Using targeted sequencing of a panel that includes 83 genes related with cerebral malformation, the authors identified homozygous splicing variants of KATNB1 gene in the first old sister which was presented in both parents as heterozygous. And the authors first analyzed the second sister using microarray and observed a partial duplication of the long arm of chr15 which is inherited from the healthy mother. With the collaboration of Italian network for undiagnosed patients, the authors performed whole exome sequencing (WES) analysis and identified homozygous frameshift variants on FAT1 gene that reported one allele in gnomAD database (4.01x106).

Overall, the genetic analysis and clinical features of both sib patients are well documented, and the authors emphasized the importance of WES as a diagnostic tool in consanguineous family study. However, the authors have to provide some additional evidence to draw the conclusion that the pathogenicity of the patients is due to the mutations each in KATNB1 and FAT1.

Minor comments:

  1. Are there any additional mutations found in the patients, especially the second patient? Authors should provide additional tables including mutations that passed the criteria and explain the reasons for exclusion, not just FAT1 gene for example.
  2. The author also should have to provide Sanger sequencing results with segregation study as supplementary data.
  3. The author also may need to provide WES quality table including mean depth of coverage.

Author Response

Thank you for your suggestions. We have added a Supplementary table S1 to provide WES quality table and additional data about depth and coverage, a Supplementary Table S2 about about rare variants of uncertain significance identified in the WES of case 2, a Supplementary Figure S2 with Sanger sequencing results with segregation study of KATNB1 and Supplementary Figure S3 with results of the FAT1 variant confirmed via Integrative Genome Viewer.

We also have added the following sentences to the discussion section to provide some additional evidence to draw the conclusion that the pathogenicity of the patients is due to the mutations each in KATNB1 and FAT1:

“In our case the family described here, the two sisters presented phenotypic pictures that were scarcely reminiscent of commonly known syndromes even though, in retrospect, the identification of homozygous variants in KATNB1 and FAT1 well explained the respective malformations. In case 1, primary microcephaly and brain malformations, including in particular nodular heterotopia, severe short stature and syndactyly, are perfectly compatible with the homozygous KATNB1 mutation we have detected [15,26]. In case 2, colobomatous-microphthalmia, ptosis, nephropathy and limb malformation correspond dramatically to what is described in the few patients with FAT1 homozygous frameshift variants reported so far [16], and strengthen the role of the novel variant we have identified.

However, even before knowing the molecular data, the malformation features of the two sisters were so diverse that the hypothesis of a single pathogenic gene variant with variable expressivity was implausible.”

Reviewer 2 Report

Peluso et al. present an interesting study of two siblings in a consanguineous family. Each of the sibling has a different phenotype and the authors identify separate gene/variants underlying the phenotype in each individual. The study is well-written and clear to follow. In particular, I like the emphasis on whole-exome sequencing as the minimum recommendation for the study of all families. It is clear that de novo variants are important in human disease more generally and that a search through all genes in the genome is critical.

Author Response

Thank you for your comment. We hope that the revisions that we made could improve the manuscript.

Reviewer 3 Report

In this article the authors present two sisters from a consanguineous family both presenting with truncating recessive mutations in two different genes belonging to rare clinical entities. While the authors do not provide functional validation of this variants, the clinical delineation is thorough. The cases are well explained and pictures are included to improve diagnosis. In general, the article is correct, well written and expands the clinical knowledge on two very rare syndromes, poorly known.

Material and methods:

Please, indicate gnomAD version or date of accession.

Results

I suggest to move web links from the results section to methodology to make it easier to follow. Some of them seem not to be working properly, at least without subscription. Bibliographical references to the bioinformatic tools would be more useful and appropriate, rather than a link.

The clinical presentation is a little bit confuse and hard to follow, may be the somatometric data can be included as a graph in each figure?

Discussion:

The discussion is very useful and the authors have done an excellent work collecting clinical data on previously described cases. To this reviewer, tables S1 and 2 should be included in the main text as they are very useful, specially in the case of FAT1, in which this work contributes to the knowledge of an emerging syndrome with sparce and limited casuistic. I also encourage the authors to mention in the discussion if the traits are absent or present (and in which degree) in the cases presented here. As an example, in line 254, in the paragraph “Microcephaly is a characteristic feature of” a mention on the presence of this trait on patient one would be useful.

I would like to also encourage the authors to include the HPOs for the clinical symptoms and, if possible, to resume Patient 1 and 2 clinical presentations in a single table coded by HPOs terms to further help on the future diagnosis of these rare syndromes. It could also be useful to include a Sup.Image with expanded pedigree of the family.

In Table 2, the protein domain affected by each mutation is mentioned, but this is not further discussed. I Think the discussion on FAT1 should be enriched. This gene is anticonstrained (Z = -0.43) for missense mutations but some missense are considered pathogenic. Also, the gene is also not constrained for LoF mutations and is included in the majority of “false positives” lists on WES analysis as it is usually mutated in a majority of individuals. Further discussion on the nature and frequency of mutations on FAT1 is needed.

As a minor comment, is recommended to avoid the use of “our” referring to the patients (line 271 as an example).

References have to be thoroughly reviewed as there are some mistakes. In line 291, Ref 13 refers do not seem the appropriate reference.

I would like to encourage the authors to include other identified VUS as a Supl. Table (specially in Case 2) as some may be acting as modificators and the inclusion of this information can improve future knowledge.

Author Response

Thank you very much for your helpful suggestions that have allowed us to improve the quality of the manuscript.

In this article the authors present two sisters from a consanguineous family both presenting with truncating recessive mutations in two different genes belonging to rare clinical entities. While the authors do not provide functional validation of this variants, the clinical delineation is thorough. The cases are well explained and pictures are included to improve diagnosis. In general, the article is correct, well written and expands the clinical knowledge on two very rare syndromes, poorly known.

Material and methods:

Please, indicate gnomAD version or date of accession.

-We have added the gnomAD version

Results

I suggest to move web links from the results section to methodology to make it easier to follow. Some of them seem not to be working properly, at least without subscription. Bibliographical references to the bioinformatic tools would be more useful and appropriate, rather than a link.

-We have moved the web links all together in the Materials and Methods paragraph

The clinical presentation is a little bit confuse and hard to follow, may be the somatometric data can be included as a graph in each figure?

-We have added to figure 1 and figure 2 the somatometric graphs about lenght, weight and head circumference to facilitate the interpretation of clinical data.

Discussion:

The discussion is very useful and the authors have done an excellent work collecting clinical data on previously described cases. To this reviewer, tables S1 and 2 should be included in the main text as they are very useful, specially in the case of FAT1, in which this work contributes to the knowledge of an emerging syndrome with sparce and limited casuistic. […] I would like to also encourage the authors to include the HPOs for the clinical symptoms and, if possible, to resume Patient 1 and 2 clinical presentations in a single table coded by HPOs terms to further help on the future diagnosis of these rare syndromes. It could also be useful to include a Sup.Image with expanded pedigree of the family.

-Relating to the table S1 and S2, unfortunately, the large tables would be very difficult to read in the in-text journal format (we already tried in the first draft and then choose to move them to the supplementary files). Adding an in-text table with the HPOs for the clinical findings in our two patients (table 1), as you suggested, and leaving the other tables as supplementary matherials seemed like a good compromise.

I also encourage the authors to mention in the discussion if the traits are absent or present (and in which degree) in the cases presented here. As an example, in line 254, in the paragraph “Microcephaly is a characteristic feature of” a mention on the presence of this trait on patient one would be useful.

We have added the following sentences to the discussion section:

“In our case the family described here, the two sisters presented phenotypic pictures that were scarcely reminiscent of commonly known syndromes even though, in retrospect, the identification of homozygous variants in KATNB1 and FAT1 well explained the respective malformations. In case 1, primary microcephaly and brain malformations, including in particular nodular heterotopia, severe short stature and syndactyly, are perfectly compatible with the homozygous KATNB1 mutation we have detected [15,26]. In case 2, colobomatous-microphthalmia, ptosis, nephropathy and limb malformation correspond dramatically to what is described in the few patients with FAT1 homozygous frameshift variants reported so far [16], and strengthen the role of the novel variant we have identified.

However, even before knowing the molecular data, the malformation features of the two sisters were so diverse that the hypothesis of a single pathogenic gene variant with varia-ble expressivity was implausible.”

In Table 2, the protein domain affected by each mutation is mentioned, but this is not further discussed. I Think the discussion on FAT1 should be enriched. This gene is anticonstrained (Z = -0.43) for missense mutations but some missense are considered pathogenic. Also, the gene is also not constrained for LoF mutations and is included in the majority of “false positives” lists on WES analysis as it is usually mutated in a majority of individuals. Further discussion on the nature and frequency of mutations on FAT1 is needed.

-We have expanded the discussion of variants in FAT1 in the discussion paragraph, distinguishing between LoF and missense variants and their associated phenotypes

As a minor comment, is recommended to avoid the use of “our” referring to the patients (line 271 as an example).

References have to be thoroughly reviewed as there are some mistakes. In line 291, Ref 13 refers do not seem the appropriate reference.

-We have revised the text and corrected any errors you have reported to us

I would like to encourage the authors to include other identified VUS as a Supl. Table (specially in Case 2) as some may be acting as modificators and the inclusion of this information can improve future knowledge.

- For case 1, no other variants of potential interest were found in the NGS panel. For case 2, we added Table S2 in the supplementary materials with other VUS variants identified in the WES.
